# RNA Sequencing Identifies WT1 Overexpression as a Predictor of Poor Outcomes in Acute Myeloid Leukemia

**DOI:** 10.3390/cancers17111818

**Published:** 2025-05-29

**Authors:** Harsh Goel, Avanish Kumar Pandey, Rahul Kumar, Rakesh Kumar, Somorjit Singh Ningombam, Farhat Naz, Harshita Makkar, Jay Singh, Shadab Ali, Anita Chopra, Amar Ranjan, Aditya Kumar Gupta, Jagdish Prasad Meena, Ganesh Kumar Viswanathan, Sameer Bakhshi, Ranjit Kumar Sahoo, Atul Batra, Goura Kishor Rath, Showket Hussain, Abhimanyu Kumar Jha, Pranay Tanwar

**Affiliations:** 1Laboratory Oncology Unit, Dr. B.R.A. Institute Rotary Cancer Hospital, All India Institute of Medical Sciences, New Delhi 110029, India; goel.harsh271@gmail.com (H.G.);; 2Division of Pediatric Oncology, Department of Pediatrics, All India Institute of Medical Sciences, New Delhi 110029, India; 3Department of Hematology, All India Institute of Medical Sciences, New Delhi 110029, India; 4Department of Medical Oncology, Dr. B.R.A. Institute Rotary Cancer Hospital, All India Institute of Medical Sciences, New Delhi 110029, India; 5Department of Radiotherapy, Dr. B.R.A. Institute Rotary Cancer Hospital, All India Institute of Medical Sciences, New Delhi 110029, India; 6Division of Molecular Oncology, National Institute of Cancer Prevention & Research, Noida 201301, India; 7Department of Biotechnology and Bioengineering, Galgotias University, Greater Noida 203201, India

**Keywords:** acute myeloid leukemia, WT1, biomarker, gene expression, prognostic marker

## Abstract

Acute Myeloid Leukemia (AML) is a cancer of the blood and bone marrow that is often associated with poor clinical outcomes, even with available treatments. Researchers are exploring the role of a gene called Wilms Tumor 1 (WT1), which may be implicated in the development and progression of AML. This study aimed to understand how WT1 gene expression can help diagnose AML, predict patient survival, and monitor how well treatments are working. By analyzing blood and BM samples from AML patients, this study discovered that elevated levels of WT1 expression at diagnosis are linked to worse outcomes. These findings indicate that measuring WT1 expression could be a valuable tool for doctors to better manage AML, guiding decisions on treatment and helping to recognize patients at higher risk for relapse.

## 1. Introduction

AML is a biologically and clinically heterogeneous hematological malignancy that originates from the myeloid lineage of the hematopoietic system. It is marked by the clonal proliferation of immature myeloid progenitor cells, due to disruptions in normal differentiation processes [1,2]. With a median diagnostic age of approximately 68 years, AML predominantly affects adults, accounting for nearly 80% of all adult leukemia diagnoses and about 18% of all leukemia cases worldwide [3]. The incidence of AML is increasing, with t-AML representing 10–15% of newly diagnosed cases. t-AML often develops as a late complication following exposure to cytotoxic therapies such as alkylating agents, topoisomerase II inhibitors, or ionizing radiation [4,5]. Both genetic predispositions and environmental factors, including prior hematological disorders like MDS and MPNs, contribute significantly to AML pathogenesis [6,7]. The classification of AML has traditionally relied on the FAB system, which is based on morphological and cytochemical characteristics, and more recently on the WHO classification, which integrates clinical, cytogenetic, and molecular data. A diagnosis of AML requires the existence of at least 20% myeloid blasts in the PB or BM [8,9]. Immunophenotyping further verifies lineage and helps subclassify AML based on antigen expression [10,11,12].

The molecular landscape of AML reflects a complex interplay between genetic mutations, epigenetic alterations, aberrant gene expression, and dysregulated hematopoiesis [13,14]. Leukemogenesis is thought to originate from oncogenic transformed hematopoietic stem or progenitor cells that acquire self-renewal capacity, forming leukemic stem cells (LSCs) [15]. LSCs sustain disease and are often resistant to standard therapies, contributing to relapse and poor outcomes [16]. Mutations in genes that regulate DNA methylation (e.g., DNMT3A, TET2, IDH1/2), proliferation, apoptosis, and differentiation underscore the heterogeneous nature of AML and its variable clinical course [17,18]. Despite advances in treatment, the prognosis for AML remains poor, with an overall five-year survival rate of approximately 29.8%. Outcomes are significantly better in younger patients, with survival rates near 68% for those under 20 years of age, but they are dramatically lower in patients over 60 years old [19,20]. Standard treatment consists of induction therapy, typically with cytarabine- and anthracycline-based regimens (the “7 + 3” protocol), followed by consolidation therapy, which may include high-dose cytarabine or AHSCT in high-risk cases [21]. However, relapse and therapy resistance continue to pose major clinical challenges, underscoring the urgent need for novel biomarkers and therapeutic targets.

The WT1 gene, situated on chromosome 11p13, encodes a zinc finger transcription factor that plays a vital role in both developmental processes and cancer biology [22]. Originally identified as a tumor suppressor in Wilms’ tumors, WT1 has also been recognized for its oncogenic role in various solid tumors and blood-related cancers [23]. It is involved in regulating key cellular functions such as apoptosis, cell growth, differentiation, and maintenance of cellular homeostasis [24]. In the hematopoietic system, WT1 expression is largely restricted to primitive CD34^+^ hematopoietic stem and progenitor cells and is minimally expressed in mature leukocytes. The aberrant expression or mutation of WT1 has been reported in conditions such as MDS and chronic myeloid leukemia (CML), often correlating with poor prognosis and therapy resistance [25]. Structurally, WT1 consists of a proline- and glutamine-rich N-terminal transcriptional regulatory domain and four C-terminal zinc finger motifs responsible for DNA binding. The gene undergoes alternative splicing to generate different isoforms (+KTS and −KTS), which serve distinct biological functions [26]. Beyond hematopoiesis, WT1 plays a role in EMT transitions during embryonic development and is implicated in maintaining tissue homeostasis in adults [27,28]. In cancers such as breast cancer, non-small cell lung cancer, and Kaposi’s sarcoma, aberrant WT1 expression has been linked to chromatin remodeling, immune evasion, and altered tumor microenvironment dynamics [29,30,31]. RNA sequencing (RNA-seq) has emerged as a transformative tool in cancer research, including hematologic malignancies like AML. By enabling the comprehensive profiling of gene expression, splice variants, and fusion transcripts, RNA-seq facilitates the identification of transcriptional programs and molecular subtypes that underlie disease pathogenesis and progression [32,33]. In AML, RNA-seq has been used to uncover biomarkers, predict therapeutic response, and explore resistance mechanisms [34,35]. The ability to analyze transcriptomic changes across different stages of the disease provides a powerful approach to dissecting leukemic heterogeneity and improving patient stratification [36].

Given its multifaceted role in hematopoiesis, leukemogenesis, and tumor progression, WT1 represents a promising candidate for biomarker discovery and therapeutic targeting in AML. Thus, this study aims to investigate WT1 gene expression profiles in AML patients at diagnosis, post-induction, and relapse stages using RT-qPCR and RNA-Seq to assess its role in disease progression and evaluate its potential as a diagnostic and prognostic biomarker.

## 2. Materials and Methods

### 2.1. Patient Samples

Between 2020 and 2024, a total of 345 BM and PB samples were prospectively collected from newly diagnosed AML patients at the following clinical stages: at diagnosis (*n* = 345), post-induction for MRD assessment (*n* = 259), and at relapse (*n* = 70). Additionally, 20 BM samples were collected as controls from individuals with benign hematologic conditions, including hypersplenism, anemia, and immune thrombocytopenic purpura. All samples were obtained from the Outpatient Department of Medical Oncology at Dr. B. R. A. IRCH, AIIMS, New Delhi, India. Informed written consent was obtained from all participants in their preferred language (English or Hindi). The study received ethical approval from the Institutional Ethics Committee of AIIMS, New Delhi (Approval No. IECPG-71/27.01.2021), in accordance with the Declaration of Helsinki.

AML diagnosis was confirmed using the morphological examination of blood and marrow smears, cytochemical staining, flow cytometry to quantify blast cells, and conventional cytogenetic analysis. Diagnosis followed the criteria set by the FAB cooperative group and the WHO. A blast count of 20% was used as the threshold for diagnosing AML, unless specific genetic abnormalities were present, in which case the threshold might differ. All enrolled AML patients were treated according to the departmental protocol at AIIMS. The initial chemotherapy regimen included daunorubicin (45–60 mg/m^2^) from days 1 to 3 and cytarabine (100 mg/m^2^) via continuous infusion from days 1 to 7. Patients who did not achieve hematologic remission received a second cycle of the same treatment, while those with inadequate responses were given high-dose chemotherapy. BM assessments were conducted on day 28 after the induction phase. CR, PR, and refractory disease were defined based on the ELN guidelines. Demographic data (age and gender) and clinical information were collected from institutional medical records and patient questionnaires. Patients were prospectively followed for five years to monitor disease progression and treatment outcomes.

Inclusion Criteria:Patients aged 18 years or older of either gender.De novo AML patients who had not received prior treatment.

Exclusion Criteria:Patients with any concurrent solid or hematologic malignancy.Individuals with a history of chemotherapy or radiotherapy.Individuals who declined informed consent or refused participation.Patients with specific hematologic conditions such as APL, Down syndrome-associated myeloid neoplasms, MDS, t-AML, or myeloid sarcoma.

### 2.2. Bone Marrow Mononuclear Cells (BMMCs) Isolation

BMMCs were isolated from AML patients and controls using Ficoll-Paque density gradient centrifugation. BM samples were diluted with PBS and layered over Ficoll-Paque (1.077 g/mL), followed by centrifugation at 300× *g* for 30 min at room temperature. The buffy coat containing BMMCs was collected, washed with PBS, and preserved in TRIzol at −80 °C for RNA extraction.

### 2.3. RNA Extraction and Quantification

Total RNA was isolated from BMMCs using TRIzol Reagent as per the manufacturer’s instructions. Following chloroform-based phase separation, RNA was precipitated with isopropanol, washed with 75% ethanol, and dissolved in RNase-free water. RNA quality and quantity were evaluated using the Qubit Fluorometer and Agilent Bioanalyzer. Samples with RIN ≥ 6.5 were selected for RNA sequencing and stored at −80 °C.

### 2.4. Discovery Phase: Comprehensive Analysis of WT1 Expression via RNA Sequencing

To explore gene expression changes associated with AML progression, we adopted a two-phase strategy. Initially, RNA-Seq was performed to identify WT1 as a gene of interest based on its differential expression between diagnosis and relapse. This discovery phase was followed by the large-scale validation of WT1 expression using RT-qPCR in an extended AML cohort. This integrative approach allowed us to confirm findings from a limited transcriptomic dataset and assess their broader clinical significance.

For RNA sequencing, total RNA was extracted from paired BM samples of five de novo and five relapsed AML patients. Only samples with RIN ≥ 6.5 were processed. Libraries were prepared using the TruSeq RNA Library Prep Kit v2, including mRNA enrichment, cDNA synthesis, and adapter ligation. Sequencing was conducted with 150 bp paired-end reads on the Illumina NovaSeq X Plus platform.

### 2.5. Library Preparation and Sequencing

RNA sequencing libraries were prepared using the TruSeq RNA Library Prep Kit v2. After poly-A selection and fragmentation, cDNA synthesis and adapter ligation were performed. The libraries were amplified, purified, and quality-checked using the Agilent Bioanalyzer before being pooled and sequenced with 150 bp paired-end reads on the Illumina NovaSeq X Plus platform.

### 2.6. Bioinformatics Analysis

RNA-Seq data were processed using our in-house bioinformatics pipeline. Quality checks were performed with FastQC and trimming with Trimmomatic. Cleaned reads were aligned to the GRCh38 genome using STAR and HISAT2, with alignment quality assessed via QualiMap. DESeq2 was used for differential expression analysis, identifying genes with log2-fold change >2 or <−2. Functional enrichment was performed using ShinyGO, KEGG, and GSEA. Protein interaction networks were built using STRING, and key hub genes were identified with cytoHubba in Cytoscape (v 3.10.3).

### 2.7. Validation of WT1 Gene Expression Profiling by Quantitative RT-PCR

To validate WT1 expression from RNA sequencing, quantitative RT-PCR was performed on 345 diagnostic, 259 post-induction (MRD), and 70 relapse-stage samples from AML patients. Total RNA (0.5–1 µg) was reverse transcribed using the Improm II RT system. Specific primers for WT1 and the reference gene GAPDH were designed using Primer-BLAST. qRT-PCR was conducted on the Bio-Rad CFX96 Real-Time System, and WT1 expression levels were quantified using the 2^−ΔΔCt^ method.

### 2.8. Statistical Analysis

Patients were divided into high and low WT1 expression groups based on ROC curve analysis. The optimal cutoff value for WT1 expression was determined using the Youden index to maximize sensitivity and specificity. The ROC curve, area under the curve (AUC), and cutoff criteria are provided in Figure A2. The Kolmogorov–Smirnov test was used to assess the normality of continuous variables. Categorical variables were compared using the Chi-square test, continuous variables with normal distribution using the Student’s *t*-test, and non-normally distributed variables using the Kruskal–Wallis test, followed by the Mann–Whitney *U* test for pairwise comparisons. For the exploratory analysis of associations between WT1 expression and clinical/hematological parameters, unadjusted *p*-values were reported due to the limited number of comparisons (six variables) and the hypothesis-generating nature of the study. Survival outcomes including OS, EFS, and DFS/RFS were estimated using the Kaplan–Meier method, and survival differences between groups were assessed using the log-rank test. OS was defined as the time from diagnosis to death from any cause, with patients alive at last follow-up censored. EFS was defined as the time from diagnosis to relapse, treatment failure, or death. DFS/RFS was calculated for patients who achieved complete remission (CR), defined by the absence of circulating blasts, the normalization of peripheral blood counts, and <5% blasts in the bone marrow. Relapse was defined by the reappearance of leukemic blasts in bone marrow, peripheral blood, or extramedullary sites. Univariate Cox regression analyses were performed to identify potential prognostic factors. Variables with *p* < 0.1 in univariate analysis were considered for inclusion in multivariate Cox regression models. The proportional hazards assumption was tested for all Cox models using Schoenfeld residuals, and no violations were observed. A *p*-value < 0.05 was considered statistically significant. All statistical analyses were conducted using GraphPad Prism version 8.0 and STATA version 11.

## 3. Results

### 3.1. Patient Characterstics

A total of 345 adult patients newly diagnosed with AML were included in this study. The cohort showed a male predominance, with 200 male patients (57.97%) and 145 female patients (42.02%). The median age at diagnosis was 40 years, ranging from 19 to 76 years. At presentation, the median total leukocyte count was 20.73 × 10^3^/µL, with a range between 0.1 and 310.4 × 10^3^/µL. The median hemoglobin level was 7.34 g/dL (range: 2.4–14.8 g/dL), and the median platelet count was 49 × 10^3^/µL, ranging from 6 to 321 × 10^3^/µL. Bone marrow examination revealed a median blast percentage of 80.7%, with values ranging from 22% to 100%. Cytochemical staining for cytoplasmic myeloperoxidase (cMPO) was performed in 276 patients, of whom 161 (58.33%) were cMPO-positive and 115 (41.66%) were cMPO-negative.

Molecular profiling using RT-PCR identified several recurrent fusion gene mutations. Among the 180 patients tested for NPM1 mutations, 29 were found to be positive. FLT3-ITD mutations were detected in 45 out of 192 patients tested, while FLT3-TKD mutations were observed in 21 out of 182 patients. CEBPA mutations were identified in 18 out of 161 patients tested. Conventional cytogenetic analysis revealed that 26 out of 152 patients were positive for the AML1-ETO fusion gene, and 10 out of 148 patients tested positive for the CBFB-MYH11 fusion. Among patients with cytogenetic results available, a normal karyotype was observed in 37 out of 116 patients, while a complex karyotype was identified in three patients, as shown in Table 1.

### 3.2. RNA Sequencing Results

RNA sequencing was performed on RNA isolated from bone marrow samples of five paired AML patients, including five de novo AML and five relapse AML cases. High-throughput paired-end sequencing generated transcriptomic profiles with an average yield of approximately 29 million raw reads per sample. The sequencing data demonstrated a median GC content of 50.43%, reflecting a balanced nucleotide composition consistent with the human transcriptome. Quality assessment indicated high reliability across all samples, with Phred quality scores showing that more than 90% of the reads in each dataset had scores above Q30, signifying high base-calling accuracy and minimal sequencing errors. The paired-end reads were uniformly distributed across samples, and the mean read length was consistently 151 base pairs. These metrics confirm the high quality and integrity of the sequencing data. Detailed sequencing statistics, including read orientation, total reads, GC content, base quality distribution, total bases generated, and mean read length, are provided in Table A1.

The quality assessment of raw RNA sequencing reads was performed using the FastQC tool, focusing on three key analytical modules, which were per base sequence quality, per base sequence content, and overrepresented sequences. The per base sequence quality analysis generated box-and-whisker plots illustrating the distribution of quality scores across each nucleotide position within the reads. In these plots, the red line indicates the median quality score, the yellow box represents the interquartile range (25th to 75th percentiles), and the whiskers denote the 10th and 90th percentiles. The blue line shows the mean quality score. The X-axis corresponds to the nucleotide position, while the Y-axis represents the Phred quality score. Background color coding highlights quality thresholds, showing green for high quality (Phred ≥ 28), orange for moderate quality (Phred 20–28), and red for low quality (Phred ≤ 20). Reads with base quality scores below the acceptable threshold (Phred < 28) were subjected to quality trimming using the Trimmomatic tool in paired-end mode to enhance overall data quality. The overrepresented sequences module identified adaptor contamination and other recurrent sequences constituting ≥1% of the reads, which were also removed during preprocessing. The efficiency of trimming is summarized in Figure 1, comparing FastQC results before and after trimming and demonstrating a marked improvement in sequence quality and the successful elimination of contaminants. After trimming and quality control, more than 95% of the cleaned reads aligned successfully with the human reference genome, confirming the high quality of the processed data for downstream bioinformatics analyses.

In this study, a total of 57,491 genes were identified across 10 cases. Of these, 19,705 genes were categorized as coding genes, while 14,229 were classified as long non-coding RNAs (lncRNAs). The distribution of reads across various gene types was also analyzed, as shown in Figure 2.

Principal component analysis (PCA) was performed as a dimensionality reduction technique to condense the attribute space and summarize the data into a smaller number of principal components. PCA clustering was used to evaluate correlations and variability among the samples. The resulting PCA plot, as shown in Figure 3, provides a visual representation of sample distribution, revealing underlying patterns and relationships between the different AML cases.

A total of 1581 genes exhibited significant differential expression (*p* < 0.05). Among these, 901 genes were found to be upregulated, while 680 genes were downregulated, as shown in Figure 4.

A heatmap was generated using log2-fold change values for all 10 samples to visually represent the differential expression patterns of genes across the different patient groups, as shown in Figure 5. This heatmap provides an overview of the gene expression variability and clustering within the Denovo-AML and Relapse-AML patient samples.

The top five downregulated genes (log2-fold change < −2, *p* < 0.05) in the analysis were PRG3, ZYG11A, COL15A1, MOCS1, and CAMP. Conversely, the top five upregulated genes (log2-fold change > 2, *p* < 0.05) were GNRHR, EPGN, GALNT9, SLC1A3, and TMPRSS11D, as shown in Figure 6.

Functional enrichment analysis revealed that differentially expressed genes (DEGs) were mainly involved in SRP-dependent co-translational protein targeting, translation initiation, and protein targeting to the endoplasmic reticulum (ER). Cellular components enriched in DEGs included cytosolic ribosomes, cell-substrate junctions, and focal adhesions. Molecular functions were enriched in cadherin binding, DNA binding, and protein kinase activity, as seen in Figure 7.

Pathway analysis was performed using GSEA to identify significant pathways associated with DEGs between Denovo-AML and Relapse-AML patients. The analysis utilized the KEGG pathway database, with a significance threshold of an adjusted *p*-value (FDR) < 0.05. The top significantly downregulated pathways included ribosome (NES = −0.6827, adj. *p*-value = 1.5 × 10^−16^), coronavirus disease-COVID-19 (NES = −0.6165, adj. *p*-value = 9.5 × 10^−16^), diabetic cardiomyopathy (NES = −0.5134, adj. *p*-value = 2.5 × 10^−7^), and oxidative phosphorylation (NES = −0.5593, adj. *p*-value = 2.3 × 10^−5^). Additional downregulated pathways included viral myocarditis, DNA replication, primary immunodeficiency, cardiac muscle contraction, and B cell receptor signaling pathway, all of which exhibited significantly negative enrichment. The only upregulated pathway identified was retinol metabolism (NES = 0.6895, adj. *p*-value = 3.5 × 10^−2^), indicating increased activity in this pathway in Denovo-AML patients, as shown in Table 2 and Figure 8.

To explore the molecular interactions among the DEGs, a PPI network was constructed. The resulting network comprised 743 nodes and 5120 edges, reflecting a high degree of connectivity and interaction among the DEGs. This extensive interconnectivity suggests the involvement of these genes in coordinated biological processes relevant to AML progression, as shown in Figure A1. Additionally, hub gene analysis was performed using the CytoHubba plugin in Cytoscape, leading to the identification of 19 hub genes with the highest degree of centrality and potential biological significance in the network, as shown in Figure 9. These hub genes may serve as key regulators and potential biomarkers for distinguishing between de novo and relapsed AML cases.

### 3.3. Validation Result of WT1 Gene Expression Using qRT-PCR

RNA sequencing was initially performed on paired BM samples from five de novo and five relapsed AML patients to identify differentially expressed genes. Among these, WT1 was notably overexpressed in de novo cases compared to relapse samples, as illustrated in Figure A3. To validate and extend these findings, WT1 gene expression was quantitatively measured using qRT-PCR in a larger, independent cohort of AML patients (Figure 10). This cohort included diagnostic (de novo) cases (*n* = 345), post-induction therapy cases (*n* = 259), and relapse cases (*n* = 70). The qRT-PCR data confirmed the RNA-Seq results, demonstrating that WT1 expression is significantly elevated in diagnostic cases relative to post-induction and relapse cases. Figure 10 illustrates WT1 expression across the three clinical timepoints using box plots. Median WT1 expression was highest in diagnostic cases, decreased substantially after induction therapy, and remained comparatively lower in relapse samples. The statistical comparison indicates a highly significant difference in expression levels between diagnostic and relapse cases (*p* < 0.001), suggesting that WT1 expression dynamically reflects disease burden and treatment response. These findings support the potential clinical utility of WT1 as a biomarker for monitoring disease status in AML, particularly in distinguishing newly diagnosed patients from those in remission or relapse.

### 3.4. Association of WT1 mRNA Expression with Clinical and Hematological Parameters in AML Patients

Patients were divided into two groups based on WT1 expression, which were WT1 low (*n* = 202) and WT1 high (*n* = 143). No significant differences were observed in age (*p* = 0.82) or sex (*p* = 0.31) between the groups. A borderline significant difference was found in blast percentage (*p* = 0.05), with more patients in the WT1 high group having blast counts ≥80%. A higher proportion of WT1 high patients had a total leukocyte count (TLC) ≥50 × 10^3^/µL, but this was not statistically significant (*p* = 0.08). A significant association was found between WT1 expression and hemoglobin levels (*p* = 0.0001), with more WT1 high patients having Hb levels < 8 g/dL. No significant difference was observed in platelet counts (*p* = 0.23). These results are summarized in Table 3.

### 3.5. Prognostic Significance of WT1 Gene Expression in AML: Univariate and Multivariate Analysis

In the univariate analysis, several clinical factors were linked to EFS and OS in AML patients (Table 4). Age showed no significant association with survival. Gender was significantly associated with both EFS (HR: 1.46, *p* = 0.006) and OS (HR: 1.47, *p* = 0.007), with females at higher risk of poor outcomes. Low hemoglobin levels (<8 g/dL) were linked to worse survival (EFS HR: 0.72, *p* = 0.029; OS HR: 0.58, *p* = 0.001). Low platelet count (<50 × 10^3^/µL) was also associated with poorer survival (EFS HR: 0.58, *p* < 0.0001; OS HR: 0.73, *p* = 0.025). High WT1 expression was significantly associated with worse OS (HR: 1.33, *p* = 0.037) but showed a borderline association with EFS (HR: 1.20, *p* = 0.056), suggesting that it may be a prognostic marker for poor survival.

In the multivariate analysis, several prognostic factors were identified (Table 5). Gender remained significant for EFS (HR: 1.62, *p* = 0.028), with females at higher risk, and approached significance for OS (HR: 1.52, *p* = 0.089). Low hemoglobin levels were strongly linked to poorer survival (EFS HR: 0.42, *p* < 0.0001; OS HR: 0.26, *p* = 0.001). Low platelet count impacted OS (HR: 0.79, *p* = 0.363) but not EFS. However, high WT1 expression did not independently predict EFS (HR: 1.07, *p* = 0.777) or OS (HR: 1.20, *p* = 0.472), indicating that its association with poor survival in the univariate analysis may be confounded by other clinical factors.

### 3.6. Impact of WT1 Gene Expression on Survival Outcomes in AML Patients

The prognostic relevance of WT1 gene expression was evaluated in relation to survival outcomes in patients with AML. The Kaplan–Meier survival analysis demonstrated that higher WT1 mRNA expression levels were associated with inferior OS and EFS. Patients classified in the WT1 high expression group showed a significantly reduced median OS and EFS compared to those in the WT1 low expression group. These findings suggest that elevated WT1 expression serves as a negative prognostic indicator in AML. The survival curves illustrating these associations are presented in Figure 11.

## 4. Discussion

AML is a complex hematological malignancy marked by the clonal expansion and impaired differentiation of immature myeloid cells. Among the numerous molecular players implicated in AML pathogenesis, the WT1 gene has emerged as a complex and multifaceted regulator. Depending on the cellular context, WT1 can function as a tumor suppressor, oncogene, transcriptional repressor, or post-transcriptional regulator. Its pleiotropic roles are underpinned by zinc finger domains located in its C-terminal region, enabling it to act as a potent transcription factor that modulates genes central to cell proliferation and metabolism [37]. WT1 may either activate or repress specific target genes, with its regulatory outcomes influenced by expression levels, isoform variants, transcriptional start sites, and the cellular environment. This nuanced regulation, together with the existence of multiple isoforms, greatly contributes to its functional complexity [38,39]. In normal hematopoiesis, WT1 is thought to play a tumor suppressor role. Its overexpression in early BM cells has been shown to inhibit growth and reduce colony-forming capacity. Under physiological conditions, WT1 expression in the BM is minimal and largely confined to primitive CD34+ hematopoietic progenitor cells [25]. However, in contrast to this suppressive role, WT1 is marked upregulated in the BM and PB of leukemia patients, highlighting its oncogenic potential in pathological settings [40]. Additionally, WT1 is also essential for mesenchymal tissue homeostasis, primarily through its participation in the Wnt4 signaling cascade. Its expression is tightly regulated, with upregulation observed in early myeloid progenitors and subsequent downregulation during terminal differentiation. Notably, both somatic mutations and the aberrant overexpression of WT1 have been recurrently observed in hematologic malignancies, frequently aligning with specific molecular subtypes and clinical features [41].

Numerous studies have underscored the prognostic significance of WT1 expression, independent of mutational status. To explore this, we performed RNA-seq on paired diagnostic and post-induction samples. Our results revealed significant WT1 upregulation at diagnosis, with marked downregulation following induction therapy, supporting its role in leukemogenesis and its potential as a biomarker for therapeutic response. These findings were further validated by qPCR, confirming elevated WT1 expression in AML patients. Our data align with a growing body of literature highlighting WT1 as a prognostic marker. Miwa et al. were among the first to report WT1 overexpression in acute leukemias, with subsequent validation in ALL, CML, MDS, and pediatric AML [40]. Consistent with our results, several studies have shown that high WT1 expression is associated with inferior clinical outcomes. For instance, the initial report highlighting its prognostic impact observed a 91% remission induction rate in patients with low WT1 levels compared to 0% in those with high WT1 expression [42]. Galimberti et al. similarly linked high WT1 expression to disease progression risk [43]. Nomdedeu et al. categorized AML patients into three prognostic groups based on WT1 transcript levels, demonstrating that those with >170 copies post-induction and >100 copies post-intensification had significantly poorer survival outcomes [44]. Brieger et al. reported WT1 overexpression in 79% of AML cases at diagnosis, with expression loss during remission and reappearance prior to relapse strongly linking WT1 to disease dynamics [45].

In our study, WT1 overexpression was associated with adverse clinical features, including elevated WBC count, low hemoglobin, and high blast percentage, while prior studies have consistently reported WT1 as an independent predictor of poor OS and EFS. Ahmad EI et al. found WT1 overexpression in 73.8% of patients and linked it to reduced remission rates and shorter OS and EFS [46]. Xu et al. reported WT1 overexpression in 92.4% of 437 AML cases, correlating with lower CR rates and poorer relapse-free survival [47]. Our multivariate Cox regression analysis indicated that WT1 expression did not independently predict survival outcomes (EFS HR: 1.07, *p* = 0.777; OS HR: 1.20, *p* = 0.472). This loss of significance in the multivariate analysis suggests that the prognostic impact of WT1 may be confounded by other clinical variables, such as anemia and gender, which showed significant associations with survival in our cohort. Bergmann et al. found WT1 mRNA present in 77% of diagnostic and relapse samples, with expression loss during remission and reappearance signaling relapse [48]. Lapillone et al. demonstrated that WT1 transcript levels above 50 × 10^4^ ABL copies post-induction were independent predictors of relapse and death [49]. Woehlecke et al. similarly linked high WT1 levels to increased relapse risk and shorter OS and EFS [50]. Weisser et al. reported that a >2-log WT1 reduction within 61–180 days of treatment correlated with better outcomes [51], and Cilloni et al. confirmed that failure to achieve this reduction predicted increased relapse risk (*p* = 0.004) [52]. Barragan et al. also observed significantly elevated WT1 levels in AML compared to controls [53], while Mehralizadeh et al. noted WT1 downregulation post-chemotherapy, with lower expression levels correlating with complete remission [54]. Collectively, these findings consistently demonstrate strong correlations between WT1 overexpression and reduced remission rates, increased risk of relapse, and decreased OS and EFS.

Given its widespread expression, WT1 has become a key marker for MRD monitoring in AML. Many studies support its utility as a marker for MRD. For instance, Ahmed et al. found WT1 overexpression in 76.7% of pediatric AML cases, particularly the FAB M4 subtype, and confirmed its prognostic value [55]. Liu et al. observed lower WT1 levels in remission compared to early stage or relapsed AML, supporting its use in MRD surveillance. Lovvik et al. demonstrated that 66% of AML patients had >20-fold WT1 overexpression at diagnosis and confirmed its utility in post-treatment monitoring [56]. Ostergaard et al. further validated WT1 as a reliable MRD tool, finding high WT1 expression in 89% of newly diagnosed patients and correlating its rise with relapse in longitudinal analysis [57]. Pozzi et al. also associated raised WT1 levels at diagnosis and post-induction with poor outcomes in de novo AML, affirming WT1 as a strong predictor of relapse and survival [58]. Weisser et al. analyzed WT1 expression in 569 samples and linked high levels to shorter OS and EFS, concluding that WT1 is effective for MRD studies and prognostication [51]. In our study, WT1 overexpression was closely linked to adverse clinical outcomes and demonstrated strong potential as an MRD marker. MRD monitoring is essential in AML for guiding post-remission therapy, determining the need for allogeneic stem cell transplantation (Allo-SCT) and enabling early relapse detection. While molecular monitoring via specific mutations (e.g., RUNX1-RUNX1T1, CBFB-MYH11, NPM1, and CEBPA) is highly sensitive, more than 50% of AML patients lack these markers. Thus, alternative markers like WT1, which are broadly applicable, are critical. Our findings support this approach, confirming WT1 as a valuable prognostic and monitoring tool in AML. The *WT1* gene undergoes alternative splicing, producing multiple isoforms, most notably those that differ by the inclusion (+KTS) or exclusion (−KTS) of the following three amino acids: lysine, threonine, and serine, positioned between zinc finger domains 3 and 4. These isoforms perform distinct biological functions; the +KTS variant is primarily involved in RNA processing and is localized to nuclear speckles, while the −KTS isoform mainly functions as a transcription factor modulating gene expression [59]. Although these isoforms may have differential roles in leukemogenesis, our study did not distinguish between +KTS and −KTS variants during expression profiling. Future studies focusing on the individual expression patterns and functional relevance of these isoforms are necessary to better understand their contributions to AML development and their utility as diagnostic, prognostic, or therapeutic targets. Given the high expression of *WT1* in AML and its limited expression in most normal adult tissues, the gene has become a prominent candidate for targeted therapies. Several therapeutic approaches are under active clinical investigation, including peptide-based vaccines, antisense oligonucleotides, and engineered T-cell therapies. One of the promising vaccine candidates, galinpepimut-S, has shown the ability to induce WT1-specific T-cell responses and was well tolerated in a Phase II clinical trial (NCT02648490) involving AML patients in remission, indicating its potential in preventing disease relapse [60]. In parallel, antisense oligonucleotides (ASOs) designed to suppress *WT1* expression have shown preclinical efficacy by triggering apoptosis in leukemic cells with elevated *WT1* levels [61]. Furthermore, T cells genetically modified to express T-cell receptors (TCRs) specific to *WT1*-derived peptide-HLA complexes are being tested in early-phase trials and offer a precise immunotherapeutic strategy [62]. While these targeted therapies are promising, continued research is essential to improve delivery mechanisms, enhance immune activation, and overcome potential resistance. Integrating WT1-based therapies into standard AML treatment regimens and identifying predictive biomarkers for therapeutic response remain important future directions.

## 5. Conclusions

This study provides strong evidence that WT1 gene expression plays a crucial role in the pathogenesis, prognosis, and treatment monitoring of AML. Through RNA sequencing and qPCR validation, WT1 was found to be consistently overexpressed at diagnosis, with expression levels significantly declining after induction therapy. Persistent high WT1 expression after treatment was associated with poor therapeutic response and an increased risk of relapse, supporting its utility as a marker for MRD monitoring. Moreover, WT1 overexpression correlated with adverse clinical features, including elevated leukocyte counts, lower hemoglobin levels, and higher bone marrow blast percentages, highlighting its association with more aggressive disease characteristics. Survival analyses further confirmed that high WT1 expression is an independent predictor of inferior OS and EFS. These findings establish WT1 expression as a valuable diagnostic, prognostic, and MRD biomarker in AML. Monitoring WT1 levels could significantly enhance risk stratification and guide therapeutic decisions. This study lays the groundwork for future research into WT1-targeted therapies and supports the integration of WT1 expression analysis into routine clinical practice for AML patients.

## Figures and Tables

**Figure 1 cancers-17-01818-f001:**
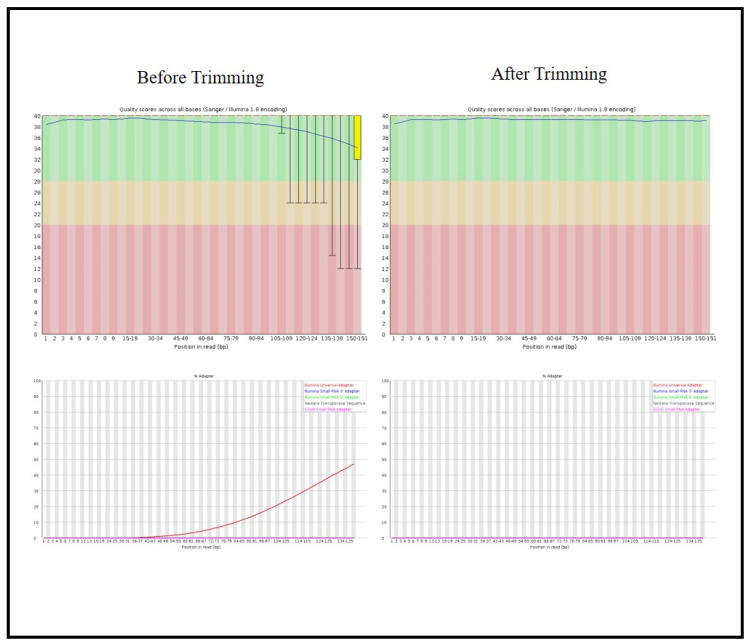
Comparison of FastQC results before and after trimming. Trimming refers to the process of removing low-quality bases and adapter sequences from raw sequencing data to improve overall data quality and accuracy for downstream analyses.

**Figure 2 cancers-17-01818-f002:**
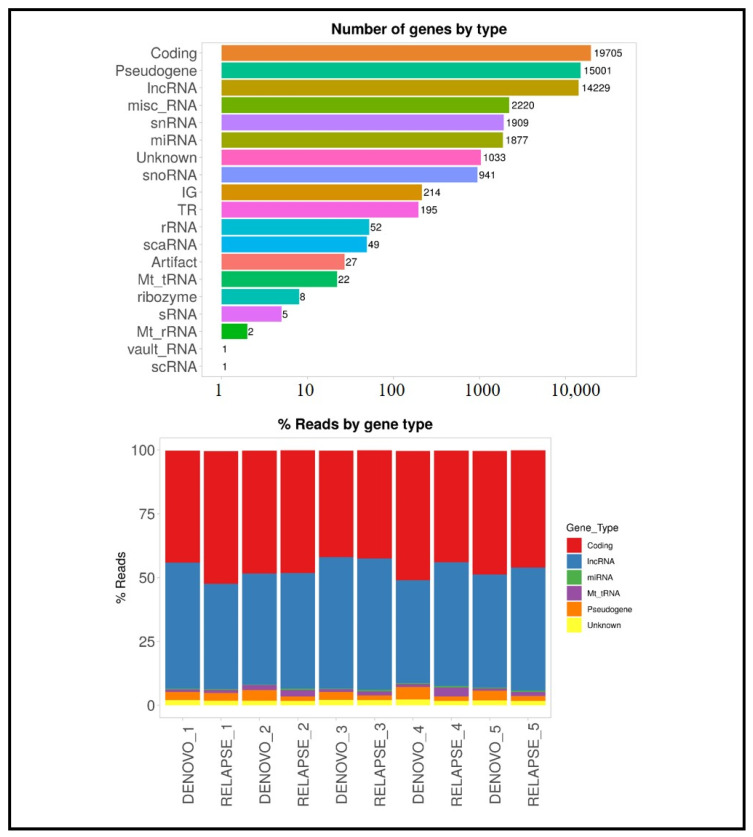
Distribution of gene types in AML Samples. Coding genes, long non-coding RNAs (lncRNAs), and other gene categories are represented as proportions of the total genes detected.

**Figure 3 cancers-17-01818-f003:**
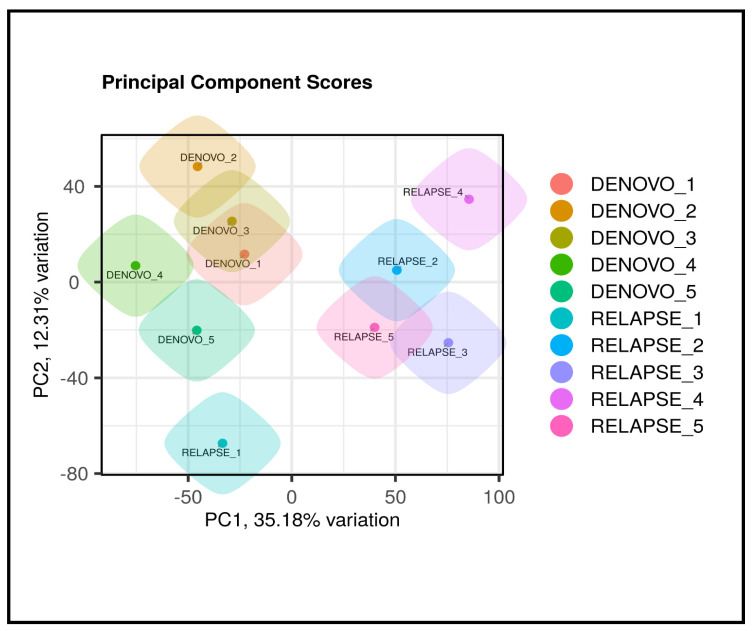
Principal component analysis (PCA) plot showing the distribution of AML samples.

**Figure 4 cancers-17-01818-f004:**
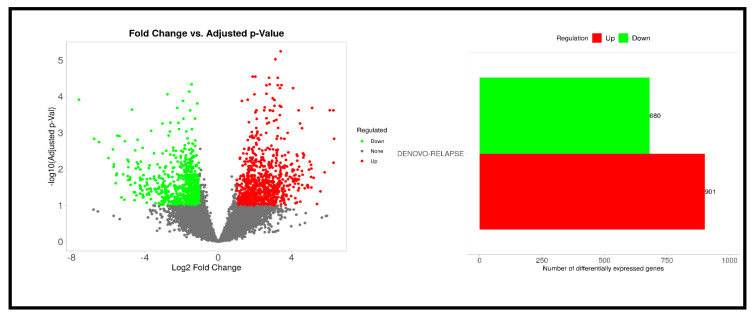
Differentially expressed genes between Denovo-AML and Relapse-AML patient samples.

**Figure 5 cancers-17-01818-f005:**
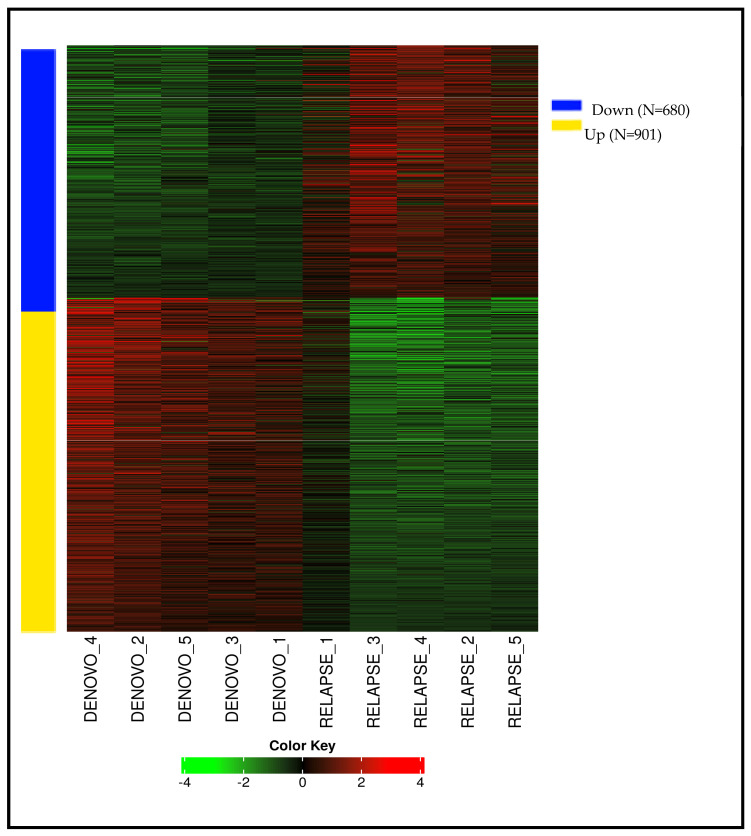
Heatmap of differential gene expression across Denovo-AML and Relapse-AML patient samples.

**Figure 6 cancers-17-01818-f006:**
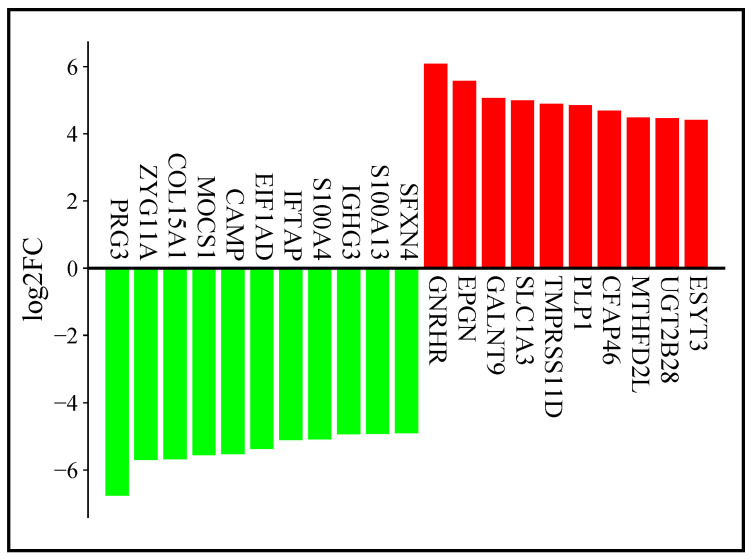
Top differentially expressed genes in AML.

**Figure 7 cancers-17-01818-f007:**
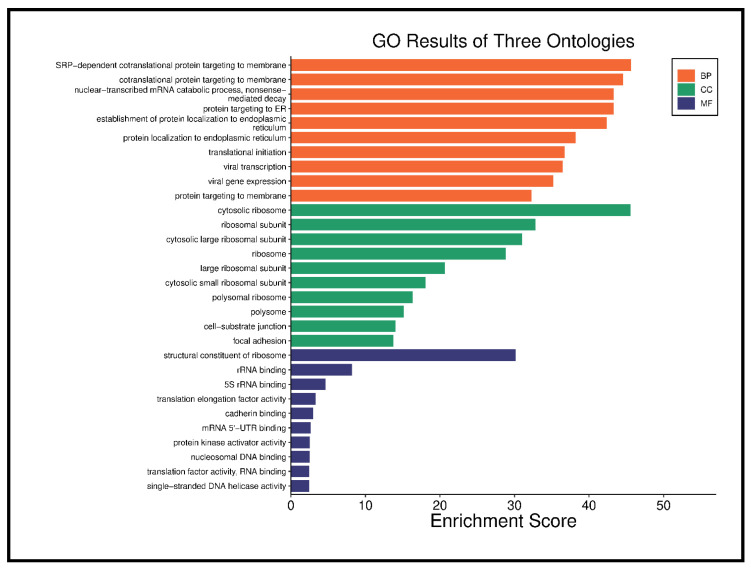
Functional enrichment analysis of significant DEGs.

**Figure 8 cancers-17-01818-f008:**
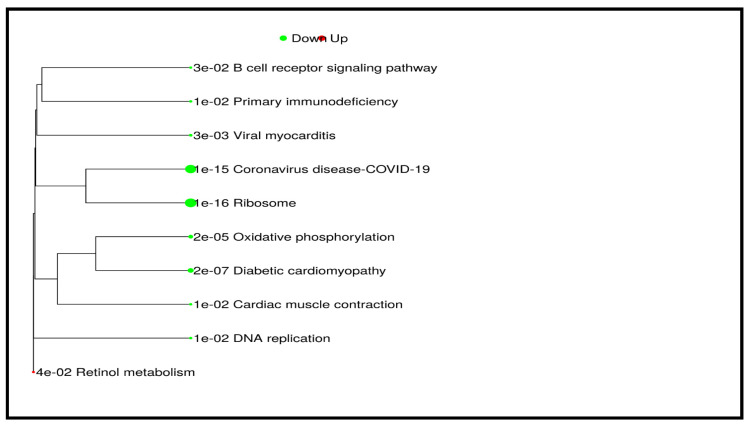
GSEA pathway analysis of DEGs in AML patients.

**Figure 9 cancers-17-01818-f009:**
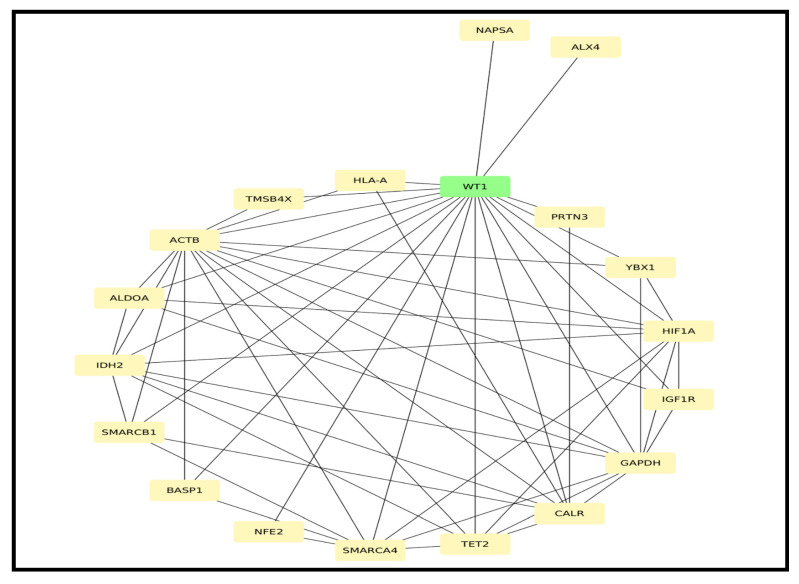
Identification of hub genes from the PPI network using Cytoscape.

**Figure 10 cancers-17-01818-f010:**
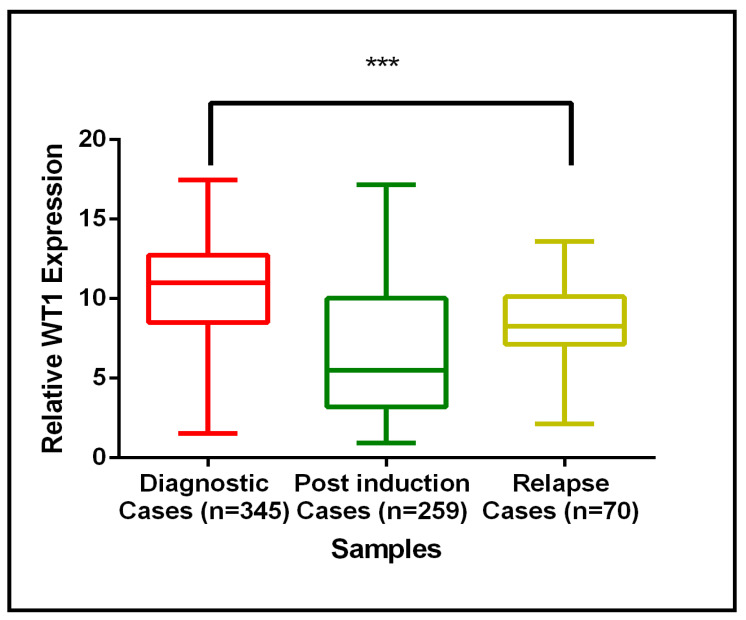
Validation of WT1 gene expression in de novo and relapse AML cases using qRT-PCR (*** indicates statistical significance at *p* < 0.001.).

**Figure 11 cancers-17-01818-f011:**
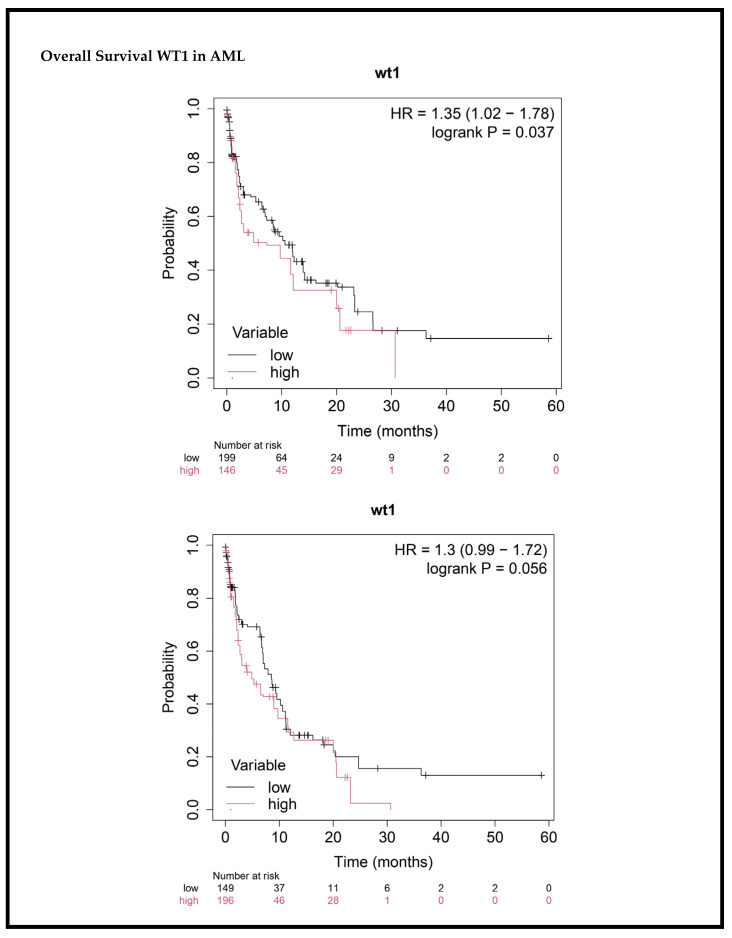
Impact of WT1 expression on overall and event-free survival in AML patients.

**Table 1 cancers-17-01818-t001:** Baseline clinical, hematological, and molecular characteristics of adult de novo AML patients (*n* = 345).

Gender Distribution	Male 200 (57.97)%		Female 145 (42.02)%	
Median age in years (range)	40 (19–76)			
Hemogram Details of Enrolled Patients				
Median Total Leukocytes Count (×10^3^/µL)	20.73 (0–310.4)			
Median Hemoglobin (g/dL) (range)	7.34 (2.4–14.8)			
Median Platelets (×10^3^/µL)	49.0 (6–321)			
Median BM Blast Percentage (%)	80.7 (22–100)			
cMPO Selection (*n* = 276)	Positive 161 (58.33)%		Negative 115 (41.66)%	
Fusion Gene Mutation Analysis by RT-PCR				
	NPM1 (*n* = 180)	FLT3-ITD (*n* = 192)	FLT3-TKD (*n* = 182)	CEBPA (*n* = 161)
Positive	29	45	21	18
Negative	151	147	161	143
Karyotype Report				
	AML-ETO (*n* = 152)	CBFB-MYH (*n* = 148)	Normal karyotype (*n* = 116)	Complex Karyotype (*n* = 116)
Positive	26	10	37	3
Negative	125	138	79	113

**Table 2 cancers-17-01818-t002:** GSEA pathway analysis of DEGs in de novo vs. relapse AML patients.

Direction	GSEA Analysis: DENOVO vs. RELAPSE	NES	Genes	Adj. *p*-Value
Down	Ribosome	−0.6827	87	1.5 × 10^−16^
	Coronavirus disease-COVID-19	−0.6165	118	9.5 × 10^−16^
	Diabetic cardiomyopathy	−0.5134	103	2.5 × 10^−7^
	Oxidative phosphorylation	−0.5593	60	2.3 × 10^−5^
	Viral myocarditis	−0.7306	15	2.8 × 10^−3^
	DNA replication	−0.6316	19	1.0 × 10^−2^
	Primary immunodeficiency	−0.6811	15	1.3 × 10^−2^
	Cardiac muscle contraction	−0.4949	35	1.5 × 10^−2^
	B cell receptor signaling pathway	−0.4565	42	2.5 × 10^−2^
Up	Retinol metabolism	0.6895	14	3.5 × 10^−2^

**Table 3 cancers-17-01818-t003:** Association of WT1 mRNA Expression Levels with Clinical and Hematological Characteristics in AML Patients.

Variable	WT1 Low Expression (*n* = 202)	WT1 High Expression (*n* = 143)	*p*-Value
Age (years)			0.82
<40	105 (52.0%)	72 (50.3%)	
≥40	97 (48.0%)	71 (49.7%)	
Total Leukocyte Count (×10^3^/µL)			0.08
<50	144 (71.3%)	89 (62.2%)	
≥50	58 (28.7%)	54 (37.8%)	
Blast Percentage (%)			0.05
<80	96 (47.5%)	63 (44.1%)	
≥80	106 (52.5%)	80 (55.9%)	
Hemoglobin (g/dL)			0.0001
<8	115 (56.9%)	115 (80.4%)	
≥8	87 (43.1%)	28 (19.6%)	
Platelets (×10^3^/µL)			0.23
<50	104 (51.5%)	64 (44.8%)	
≥50	98 (48.5%)	79 (55.3%)	
Sex			0.31
Male	122 (61.4%)	78 (54.6%)	
Female	80 (39.6%)	65 (45.4%)	

**Table 4 cancers-17-01818-t004:** Univariate analysis of prognostic factors for EFS and OS in AML patients.

Variables	Event Free Survival			Overall Survival		
	HR	95% CI	*p* Value	HR	95% CI	*p* Value
Age at Diagnosis						
<40 years (*n* = 117) ≥40 years (*n* = 168)	1.057876	0.81–1.38	0.673	1.097428	0.83–1.44	0.510
Gender						
Male (*n* = 200) Female (*n* = 145)	1.455092	1.11–1.89	0.006	1.470668	1.11–1.94	0.007
Total Leukocyte Count (×10^9^/L)						
<50 (*n* = 233) ≥50 (*n* = 112)	1.043832	0.79–1.37	0.757	1.092357	0.82–1.45	0.542
BM Blast (%)						
<80% (*n* = 159) ≥80% (*n* = 186)	0.9498176	0.73–1.23	0.701	1.21451	0.92–1.60	0.169
Hemoglobin (g/dL)						
<8 (*n* = 230) ≥8 (*n* = 115)	0.7187736	0.53–0.96	0.029	0.580122	0.41–0.80	0.001
Platelets (×10^3^/µL)						
<50 (*n* = 168) ≥50 (*n* = 175)	0.5778738	0.44–0.75	0.0001	0.7278208	0.55–0.96	0.025
WT1 Expression						
High (*n* = 143) Low (*n* = 202)	1.204193	0.92–1.57	0.056	1.329035	1.00–1.76	0.037

**Table 5 cancers-17-01818-t005:** Multivariate Cox regression analysis of prognostic factors for event-free survival and overall survival in AML patients.

Variables	Event Free Survival			Overall Survival		
	HR	95% CI	*p* Value	HR	95% CI	*p* Value
Age at Diagnosis						
<40 years (*n* = 117) ≥40 years (*n* = 168)	0.9420834	0.62–1.41	0.775	0.9210855	0.57–1.46	0.728
Gender						
Male (*n* = 200) Female (*n* = 145)	1.620069	1.05–2.48	0.028	1.515567	0.93–2.44	0.089
Total Leukocyte Count (×10^9^/L)						
<50 (*n* = 233) ≥50 (*n* = 112)	1.371956	0.83–2.26	0.217	1.457944	0.87–2.42	0.148
BM Blast (%)						
<80% (*n* = 159) ≥80% (*n* = 186)	0.9102344	0.57–1.44	0.689	1.181538	0.75–1.84	0.465
Hemoglobin (g/dL)						
<8 (*n* = 230) ≥8 (*n* = 115)	0.4168227	0.26–0.64	0.0001	0.2639707	0.15–0.45	0.001
Platelets (×10^3^/µL)						
<50 (*n* = 168) ≥50 (*n* = 175)	0.9836502	0.63–1.51	0.940	0.7934608	0.48–1.30	0.363
WT1 Expression						
High (*n* = 143) Low (*n* = 202)	1.070796	0.66–1.71	0.777	1.202246	0.72–1.98	0.472

## Data Availability

The data generated and analyzed during this study are available from the corresponding author upon reasonable request. Due to ethical considerations and restrictions outlined by the study’s approval and the informed consent obtained from participants, access to the data is limited. However, the datasets produced in this research have been deposited in the NCBI repository under BioProject ID PRJNA1223666.

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
