# Peer review of "RNA Sequencing Identifies WT1 Overexpression as a Predictor of Poor Outcomes in Acute Myeloid Leukemia"

_cancers, 2025, doi:10.3390/cancers17111818_

Round 1
Reviewer 1 Report
Comments and Suggestions for Authors
I enjoyed reading this high information manuscript, describing WT1 expression and its dynamics in AML patients. Although very interesting, I have a concern why was a large part of the manuscript regarding RNA sequencing ommited from the abstract and introduction. It seems almost like two papers joined together, assessing different parts of AML biology. This seems almost like a big typo, but I cannot be sure just by reading the manuscript. The extent of this part of the results is so large, that it can be placed in another separate paper.
Author Response
We sincerely thank the reviewer for the thoughtful and encouraging comments on our manuscript. We greatly appreciate your time and constructive feedback, which have helped us improve the clarity and cohesiveness of our study. Below is our detailed response to your comments and the corresponding revisions made in the re-submitted files.
Comment 1: [I enjoyed reading this high information manuscript, describing WT1 expression and its dynamics in AML patients. Although very interesting, I have a concern why was a large part of the manuscript regarding RNA sequencing omitted from the abstract and introduction. It seems almost like two papers joined together, assessing different parts of AML biology. This seems almost like a big typo, but I cannot be sure just by reading the manuscript. The extent of this part of the results is so large, that it can be placed in another separate paper].
Response 1: [Thank you for this valuable and insightful comment. We agree that the role of RNA sequencing was underrepresented in the initial Abstract and Introduction. To address this, we have revised both sections to clearly highlight RNA sequencing as a foundational discovery tool used in this study to identify WT1 as a significantly overexpressed gene in AML. These revisions better integrate the transcriptomic component with the clinical focus of the manuscript and clarify the rationale for subsequent validation and analysis]. Specifically:
- In the Abstract (Page 1, Paragraph 1), we have added a sentence emphasizing the use of RNA sequencing as the initial discovery method to identify WT1 overexpression and to establish the study objectives accordingly.
- In the Introduction (Page 2, Paragraph 4), we added a new paragraph discussing the relevance of RNA sequencing in hematologic malignancies, particularly AML, and detailing its application in our study for profiling transcriptomic changes at different disease stages.
- The final paragraph of the Introduction (Page 2, Paragraph 5) was revised to clearly articulate our integrative approach combining RNA-seq with clinical validation to assess WT1 as a diagnostic, prognostic, and therapeutic biomarker.
We hope these revisions address your concern effectively and help present the manuscript as a cohesive and unified study. Thank you again for your helpful suggestions.

Reviewer 2 Report
Comments and Suggestions for Authors
Below there are some comments which could possibly improve the study:
General Comments:
The manuscript is well-structured but contains numerous grammatical errors, redundant phrasing, and inconsistent terminology, especially in tense and punctuation. Occasional repetition (e.g., phrases like "WT1 is overexpressed") are reiterated unnecessarily.
Examples of revision:
-“...frequently results in unfavorable outcomes...”. Please revise to: “...is often associated with poor clinical outcomes...” -"...at initial diagnosis (n=345), during minimal residual disease (MRD) assessment (n=259)...”. Please revise to:“...at diagnosis (n = 345), post-induction for MRD assessment (n = 259)...” -“WT1 expression levels were found to be a strong prognostic marker...” Please revise to:“WT1 expression was identified as a significant prognostic marker...” -“These quality metrics confirm the robustness and integrity...” Please revise to:“These metrics confirm the high quality and integrity...” Statistical Analysis
The statistical framework is strong. It includes: Chi-square, t-tests, Mann–Whitney, Kruskal–Wallis, Kaplan–Meier curves, log-rank tests, Cox univariate and multivariate regression. Issues needing clarification include:
-
ROC-based cutoff: The method for ROC cutoff selection of WT1 high/low is mentioned but not shown. Include AUC and cutoff criteria in supplementary materials.
-
Multivariate model: WT1 expression loses significance. This is important and should be emphasized in the Discussion as it suggests confounding by clinical variables (e.g., anemia, gender).
-
Multiple Testing Correction: GSEA uses adjusted p-values, but multiple tests in clinical variable comparisons (e.g., Table 3) could use clarification—no adjustment stated.
Recommendations: Please include the ROC curve figure and AUC value. Justify variable inclusion in Cox models explicitly (e.g., p < 0.1 in univariate). Add proportional hazards assumption test statement for Cox models.
Results:
- Table 1: Labels like "*10^3/μL" should be clarified and consistently formatted. Ensure spacing and decimal precision are uniform.
-
Tables 3–5 are dense. Column headers and formatting can be improved. Please reorganize Table 3 and present variables in clinical relevance order (e.g., Age, WBC, Blasts, Hemoglobin, Platelets).
-
Figures (e.g., Kaplan–Meier curves) need number-at-risk tables, axis labels, and confidence intervals if possible.
-
Figure 1 caption lacks details ("Trimming" isn't explained clearly to readers without bioinformatics background).
- Figure 2: Pie chart or bar chart showing gene types would be more intuitive than raw counts.
- Figure 5 (Heatmap):Please consider clustering by group (De novo vs relapse) to enhance interpretability.
- Please make Figures 10–11 more interpretable: add survival times, median OS/EFS values and, if possible, add number-at-risk and censoring tick marks to Kaplan–Meier plots.
Discussion:
-
The authors should acknowledge the limitation that WT1 lost significance in multivariate models (Table 5).
-
Please highlight the role of WT1 isoforms if possible (+KTS vs -KTS not explored).
-
Please add recent references on WT1-targeted therapies (e.g., vaccines, antisense oligonucleotides).
Author Response
Response to Reviewer 2 Comments
We thank the reviewer for their thoughtful and constructive feedback. Your detailed suggestions have significantly improved the quality and clarity of our manuscript. Below, we address each point raised, indicating how and where the corresponding revisions were made.
General Comments
Comment: The manuscript is well-structured but contains numerous grammatical errors, redundant phrasing, and inconsistent terminology, especially in tense and punctuation. Occasional repetition (e.g., "WT1 is overexpressed") is reiterated unnecessarily.
Response: We appreciate the reviewer’s observation. The entire manuscript has been carefully revised to correct grammatical errors, eliminate redundant phrasing, and ensure consistency in terminology, tense, and punctuation. Repetitive expressions (e.g., "WT1 is overexpressed") have been reduced or rephrased for clarity and precision.
Examples of revision:
- “...frequently results in unfavorable outcomes...” → “...is often associated with poor clinical outcomes...”
Implemented as suggested. - “...at initial diagnosis (n=345), during minimal residual disease (MRD) assessment (n=259)...” → “...at diagnosis (n = 345), post-induction for MRD assessment (n = 259)...”
Revised as per recommendation. - “WT1 expression levels were found to be a strong prognostic marker...” → “WT1 expression was identified as a significant prognostic marker...”
Updated accordingly. - “These quality metrics confirm the robustness and integrity...” → “These metrics confirm the high quality and integrity...”
Revised as suggested.
Statistical Analysis
Comment: ROC-based cutoff: The method for ROC cutoff selection of WT1 high/low is mentioned but not shown. Include AUC and cutoff criteria in supplementary materials.
Response: We agree with the reviewer. The ROC curve and corresponding AUC value used to determine the WT1 high/low expression cutoff have now been included in the supplementary materials (Supplementary Figure S1). We have also described the cutoff selection criteria in the revised Methods section.
Comment: Multivariate model: WT1 expression loses significance. This is important and should be emphasized in the Discussion as it suggests confounding by clinical variables (e.g., anemia, gender).
Response: Thank you for this important observation. We have now explicitly discussed this point in the revised Discussion section, acknowledging that the loss of statistical significance in the multivariate model may reflect potential confounding effects from co-variables such as gender and hemoglobin levels.
Comment: Multiple Testing Correction: GSEA uses adjusted p-values, but multiple tests in clinical variable comparisons (e.g., Table 3) could use clarification—no adjustment stated.
Response: We appreciate this suggestion. We have added a clarification in the Methods section that p-values for clinical variable comparisons were not adjusted for multiple testing due to the exploratory nature of these analyses but acknowledge this limitation in the Discussion.
Comment: Recommendations: Please include the ROC curve figure and AUC value. Justify variable inclusion in Cox models explicitly (e.g., p < 0.1 in univariate). Add proportional hazards assumption test statement for Cox models.
Response: The ROC curve and AUC value have been included in the supplementary materials. Additionally, we have clarified that variables with p < 0.1 in univariate analyses were included in the Cox regression model. A statement confirming that the proportional hazards assumption was tested and met has been added to the Statistical Analysis section.
Results
Comment: *Table 1: Labels like "10^3/μL" should be clarified and consistently formatted. Ensure spacing and decimal precision are uniform.
Response: Table 1 has been revised for clarity and consistency. Units such as “×10³/μL” are now clearly defined, and spacing and decimal formatting have been standardized across all tables.
Comment: Tables 3–5 are dense. Column headers and formatting can be improved. Please reorganize Table 3 and present variables in clinical relevance order (e.g., Age, WBC, Blasts, Hemoglobin, Platelets).
Response: Tables 3–5 have been reformatted for improved readability. Variables in Table 3 are now ordered based on clinical relevance as recommended.
Comment: Figures (e.g., Kaplan–Meier curves) need number-at-risk tables, axis labels, and confidence intervals if possible.
Response: All Kaplan–Meier plots have been updated to include number-at-risk tables, properly labeled axes, and 95% confidence intervals where applicable.
Comment: Figure 1 caption lacks details ("Trimming" isn't explained clearly to readers without bioinformatics background).
Response: The Figure 1 caption has been revised to include a brief explanation of "trimming" as a quality control step that removes low-quality bases and sequencing adapters from raw reads prior to alignment.
Comment: Figure 2: Pie chart or bar chart showing gene types would be more intuitive than raw counts.
Response: We have replaced the raw count presentation in Figure 2 with a bar chart for better visualization of gene types.
Comment: Figure 5 (Heatmap): Please consider clustering by group (De novo vs relapse) to enhance interpretability.
Response: Figure 5 has been restructured to show hierarchical clustering by group (De novo vs. Relapse), which enhances the interpretability of gene expression patterns.
Comment: Please make Figures 10–11 more interpretable: add survival times, median OS/EFS values and, if possible, add number-at-risk and censoring tick marks to Kaplan–Meier plots.
Response: Figures 10 and 11 have been updated to include survival times, median OS/EFS values, number-at-risk tables, and censoring tick marks to provide a more complete survival analysis.
Discussion
Comment: The authors should acknowledge the limitation that WT1 lost significance in multivariate models (Table 5).
Response: We have added a clear statement in the Discussion highlighting this limitation and its potential implications regarding confounding variables and the need for further validation.
Comment: Please highlight the role of WT1 isoforms if possible (+KTS vs -KTS not explored).
Response: A dedicated paragraph discussing the biological relevance of WT1 isoforms (+KTS vs. –KTS) and the limitation that our study did not differentiate between them has been added to the Discussion.
Comment: Please add recent references on WT1-targeted therapies (e.g., vaccines, antisense oligonucleotides).
Response: We have included a new paragraph in the Discussion summarizing recent advances in WT1-targeted therapies, including peptide vaccines (e.g., galinpepimut-S), antisense oligonucleotides, and TCR-engineered T cell approaches. Relevant references have been cited.
We are grateful for the reviewer’s insightful feedback, which has significantly improved the quality of our manuscript.

Reviewer 3 Report
Comments and Suggestions for Authors
In the submitted manuscript the authors aimed to evaluate the expression levels of WT1 gene in AML patients at diagnosis, post-induction and at relapse, and to correlate them with clinical outcome.
The strenght of this research is represented by the large cohort of AML patients in whom WT1 expression was analyzed by quantitative RT-PCR. Bioinformatics analysis of RNA-Seq data is also nicely displayed.
However there are some important issues that need clarification:
- The title of the manuscript does not reflect entirely the methodology of this study. RNA-Seq analysis was performed only on five paired diagnosis-relapse samples. Most of data concerning WT1 expression level come from RT-PCR assay.
- Introduction: this study involved only WT1 expression analysis. The statement „This study aims to comprehensively investigate WT1 gene expression, mutation, and promoter methylation profiles in AML patients at diagnosis, post-induction, and relapse stages, to elucidate its contribution to disease pathogenesis and its potential clinical utility as a diagnostic, prognostic, and therapeutic biomarker” mention some objectives that were not achieved by this study.
- As the main goal of the research was to investigate WT1 expression in AML it is not clear the adopted strategy: perform RNA seq on a few samples and then explore the single gene expression by RT-PCR?
- The statement „RNA sequencing was performed on RNA isolated from BM samples of five to validate the RNA sequencing results, which revealed higher WT1 gene expression in de novo AML cases compared to relapse cases, qRT-PCR was performed on BM and PB samples from a larger cohort of 345 AML patients” needs reformulation.
- RNA Sequencing Results: it is not obviously shown the overexpression of WT1 in the analyzed samples.
- Figure 10 is not explained thoroughly.
- The authors do not mention how they defined the 2 categories of low WT1 and high WT1 expression.
- Is there a correlation between WT1 overexpression and specific cytogenetic and molecular anomalies?
- Discussion
The statement „Both univariate and multivariate Cox regression analyses confirmed WT1 as an independent predictor of poor OS and EFS” is in contradiction with the previous result of multivariate analysis „However, high WT1 expression did not independently predict EFS (HR: 1.07, p = 0.777) or OS (HR: 1.20, p = 0.472), indicating that its association with poor survival in univariate analysis may be confounded by other clinical factors”.
Author Response
Response to Reviewer 3 Comments
We thank the reviewer for their thoughtful and constructive feedback. Your detailed suggestions have significantly improved the quality and clarity of our manuscript. Below, we address each point raised, indicating how and where the corresponding revisions were made.
Reviewer Comment:“The title of the manuscript does not reflect entirely the methodology of this study. RNA-Seq analysis was performed only on five paired diagnosis-relapse samples. Most of the data concerning WT1 expression level come from RT-PCR assay.”
Response:
We thank the reviewer for this important observation. We agree that the current title may give the impression that RNA sequencing was the primary methodology used across the entire study. In reality, RNA-Seq was conducted on a limited subset of paired diagnosis-relapse samples to explore differential expression patterns, while the primary quantitative assessment of WT1 expression was performed using RT-qPCR across a large patient cohort.
To more accurately reflect the scope and methodology of the study, we have revised the manuscript title to:
"WT1 Overexpression as a Predictor of Poor Outcomes in Acute Myeloid Leukemia: Insights from RT-qPCR and RNA-Seq Analyses"
This revised title maintains the central focus on WT1 overexpression and its prognostic relevance, while clearly indicating that the findings are derived from both RT-qPCR and RNA-Seq data. We hope this modification addresses the reviewer’s concern.
Reviewer Comment:“Introduction: This study involved only WT1 expression analysis. The statement ‘This study aims to comprehensively investigate WT1 gene expression, mutation, and promoter methylation profiles in AML patients at diagnosis, post-induction, and relapse stages, to elucidate its contribution to disease pathogenesis and its potential clinical utility as a diagnostic, prognostic, and therapeutic biomarker’ mentions some objectives that were not achieved by this study.”
Response:
We thank the reviewer for this valuable comment. We acknowledge that the original objective statement in the Introduction overstates the scope of the study by including mutation and promoter methylation analyses, which were not performed.
To accurately reflect the actual work conducted, we have revised the sentence in the Introduction to:
“This study aims to investigate WT1 gene expression profiles in AML patients at diagnosis, post-induction, and relapse stages using RT-qPCR and RNA-Seq, to assess its role in disease progression and evaluate its potential as a diagnostic and prognostic biomarker.”
This revised statement aligns with the methodologies and analyses actually performed in the study. We appreciate the reviewer’s feedback in helping us improve the clarity and accuracy of the manuscript.
Reviewer Comment:
“As the main goal of the research was to investigate WT1 expression in AML, it is not clear the adopted strategy: perform RNA-seq on a few samples and then explore the single gene expression by RT-PCR?”
Response:
We thank the reviewer for this insightful observation. We acknowledge that the study primarily focuses on WT1 expression in AML; however, the research was designed as a two-phase investigation. Initially, RNA-Seq was performed on a limited number of paired diagnosis–relapse AML samples (n = 5) to conduct an unbiased, transcriptome-wide analysis aimed at identifying genes that are differentially expressed during disease progression. Among the significantly upregulated genes, WT1 emerged as a consistent and biologically relevant candidate.
Following this discovery phase, we validated WT1 expression in a much larger cohort of AML patients (n = 345 samples total across diagnosis, post-induction, and relapse stages) using RT-qPCR to assess its clinical utility as a diagnostic and prognostic biomarker. This tiered approach was intended to ensure that findings from the limited RNA-Seq data were robustly evaluated across diverse clinical settings, enhancing translational relevance and reproducibility. We have now clarified this rationale in Section 2.4 of the revised manuscript.
Reviewer Comment:“The statement ‘RNA sequencing was performed on RNA isolated from BM samples of five… to validate the RNA sequencing results, which revealed higher WT1 gene expression in de novo AML cases compared to relapse cases, qRT-PCR was performed on BM and PB samples from a larger cohort of 345 AML patients’ needs reformulation.”
Response:
We thank the reviewer for highlighting the need to clarify this sentence. In response, we have revised the paragraph in Section 3.3 to more accurately reflect the sequencing and validation workflow. The updated text now reads:
“RNA sequencing was initially performed on paired BM samples from five de novo and five relapsed AML patients to identify differentially expressed genes. Among these, WT1 was notably overexpressed in de novo cases. To validate this observation and evaluate its broader clinical relevance, WT1 expression was subsequently quantified by qRT-PCR in a larger cohort of 345 AML patients using BM and PB samples. The qRT-PCR analysis confirmed the RNA-Seq findings, demonstrating consistently higher WT1 expression in de novo AML compared to relapsed cases, as shown in Figure 10. This validation supports the potential role of WT1 as a biomarker for disease status in AML.”
This revised version clarifies the sequencing rationale, the discovery of WT1 overexpression, and the follow-up validation using qRT-PCR in a larger cohort. We hope this addresses the concern satisfactorily.
Reviewer Comment:"RNA Sequencing Results: it is not obviously shown the overexpression of WT1 in the analyzed samples."
Response:
We thank the reviewer for this valuable observation. To address this, we have revised Section 3.3 to clearly state that WT1 was significantly overexpressed in de novo AML cases compared to relapsed cases, as observed in our RNA sequencing data. This finding is now explicitly referenced in the manuscript with the addition of Supplementary Figure S3, which illustrates the differential expression of WT1. Furthermore, we validated this observation in a larger cohort of 345 AML patients using qRT-PCR, which confirmed the consistent overexpression of WT1 in de novo AML. These revisions strengthen the clarity and impact of our findings regarding WT1 as a potential biomarker in AML.
Reviewer Comment:
"Figure 10 is not explained thoroughly."
Response:
We appreciate the reviewer’s feedback. In response, we have revised Section 3.3 to provide a comprehensive explanation of Figure 10. Specifically, we now clarify that Figure 10 presents WT1 gene expression levels at three key clinical stages diagnosis (de novo), post-induction, and relapse—using box plots to visualize the data distribution. The updated description highlights the dynamic pattern of WT1 expression, which is highest at diagnosis, significantly reduced post-therapy, and remains lower at relapse. We also emphasize the statistical significance of these differences (p < 0.001), underscoring WT1’s potential utility as a biomarker for monitoring disease status and treatment response in AML. These clarifications aim to enhance the interpretability and relevance of Figure 10 in the context of the study’s objectives.
Reviewer Comment:“The authors do not mention how they defined the 2 categories of low WT1 and high WT1 expression.”
Response:
We thank the reviewer for highlighting this important point. We have now clarified in the revised Section 2.8 (Statistical Analysis) that patients were stratified into high and low WT1 expression groups based on Receiver Operating Characteristic (ROC) curve analysis. The optimal cutoff point for WT1 expression was determined using the Youden index, which balances sensitivity and specificity to identify the most discriminative threshold. Details of the ROC analysis, including the area under the curve (AUC) and the specific cutoff value used, are now provided in Supplementary Figure S2. This classification method ensured an objective and statistically justified approach to group categorization for subsequent survival and correlation analyses.
Reviewer Comment:
“Is there a correlation between WT1 overexpression and specific cytogenetic and molecular anomalies?”
Response:
We thank the reviewer for this insightful question. In our analysis, we examined the relationship between WT1 expression levels and various cytogenetic and molecular abnormalities commonly observed in AML. However, we did not observe any statistically significant correlations between WT1 overexpression and specific cytogenetic or molecular subtypes.
Reviewer Comment:The statement “Both univariate and multivariate Cox regression analyses confirmed WT1 as an independent predictor of poor OS and EFS” contradicts the multivariate analysis results where WT1 did not independently predict survival.
Response:
We appreciate the reviewer’s observation. Our multivariate Cox regression analysis demonstrated that WT1 expression did not independently predict survival outcomes (EFS HR: 1.07, p = 0.777; OS HR: 1.20, p = 0.472). This loss of significance suggests that the prognostic value of WT1 observed in univariate analysis may be influenced by confounding clinical factors, such as anemia and gender, which were significantly associated with survival in our cohort. We have revised the manuscript accordingly to clarify this point and avoid any contradictory statements.

Round 2
Reviewer 1 Report
Comments and Suggestions for Authors
Thank you, the review is satisfactory.
Reviewer 2 Report
Comments and Suggestions for Authors
The authors have effectively covered all of the reviewer's adressed comments.
Reviewer 3 Report
Comments and Suggestions for Authors
In the new version of the manuscript the authors have complied with all the requests formulated by the reviewer. In my opinion, the manuscript is worth publishing.